# HIV-MTB Co-Infection Reduces CD4+ T Cells and Affects Granuloma Integrity

**DOI:** 10.3390/v16081335

**Published:** 2024-08-21

**Authors:** Suyue Huang, Maoying Liu, Hui Zhang, Wei Song, Wenjuan Guo, Yanling Feng, Xin Ma, Xia Shi, Jianjian Liu, Li Liu, Tangkai Qi, Zhenyan Wang, Bo Yan, Yinzhong Shen

**Affiliations:** 1Shanghai Public Health Clinical Center, Fudan University, Shanghai 201508, China; bedelia1234@163.com (S.H.); songwei@shaphc.org (W.S.);; 2Department of Microbiology, School of Basic Medical Sciences, Guizhou Medical University, Guiyang 550025, China; 3Department of Ultrasound, Zhongshan Hospital, Fudan University, Shanghai 200032, China

**Keywords:** HIV, tuberculosis, granuloma, co-infection, CD4+ T cell

## Abstract

Granuloma is a crucial pathological feature of tuberculosis (TB). The relationship between CD4+ T cells in both peripheral blood and granulomatous tissue, and the integrity of granulomas in Human Immunodeficiency Virus (HIV)–MTB co-infection, remains unexplored. This study collected biopsy specimens from 102 TB patients (53 with HIV-MTB co-infection and 49 only with TB). Hematoxylin and eosin (HE) staining and immunohistochemical staining were performed, followed by microscopic examination of the integrity of tuberculous granulomas. Through statistical analysis of peripheral blood CD4+ T cell counts, tissue CD4+ T cell proportion, and the integrity of granulomas, it was observed that HIV infection leads to poor formation of tuberculous granulomas. Peripheral blood CD4+ T cell counts were positively correlated with granuloma integrity, and there was a similar positive correlation between tissue CD4+ T cell proportions and granuloma integrity. Additionally, a positive correlation was found between peripheral blood CD4+ T cell counts and the proportion of CD4+ T cells in granuloma tissues. Therefore, HIV infection could impact the morphology and structure of tuberculous granulomas, with a reduced proportion of both peripheral blood and tissue CD4+ T lymphocytes.

## 1. Introduction

HIV and *Mycobacterium tuberculosis* (MTB) are both major pathogens that pose a threat to human health. In July 2023, the United Nations Acquired Immunodeficiency Syndrome (AIDS) program (UNAIDS) released data showing that as of the end of 2022, there were approximately 39 million people living with HIV globally; in 2022, there were 1.3 million new HIV infections, and 630,000 people died from AIDS-related diseases [1]. According to the Global TB Report 2023, published by the World Health Organization (WHO), there were 10.6 million new TB cases worldwide, with 6.7% of these patients co-infected with HIV [2]. TB remains the leading cause of death in people infected with HIV. In 2022, 1.3 million people died from TB, including 167,000 HIV-infected individuals [2].

Studies show that in HIV-MTB co-infection, there is a complex interaction between HIV and MTB that influences the incidence and progression of the disease. HIV infection leads to immunodeficiency, which promotes the infection, dissemination, and disease development of MTB; the risk of latent tuberculosis infection (LTBI) progressing to active tuberculosis (ATB) also significantly increases. Additionally, co-infected patients have an increased lung burden of MTB, leading to disseminated TB [3]. Meanwhile, MTB infection can enhance HIV transcription and replication, increase variability, and may increase the susceptibility of immune cells to HIV. The associated immune microenvironment disorder can also exacerbate the condition [4,5]. Thus, the interaction between HIV and MTB makes the condition of co-infected patients relatively more complex and severe, presenting greater challenges for the diagnosis and treatment of TB in individuals with AIDS.

Due to the functional impairment of macrophages and dendritic cells [6,7,8] and the depletion of CD4+ T lymphocytes caused by HIV infection, the trajectory of granulomas may be affected by HIV infection. A systematic review and meta-analysis in 2016 suggested that the current findings on how HIV infection affects the formation of tuberculous granulomas were contradictory [9]. For example, among the 13 selected papers, some believed that peripheral blood CD4+ T lymphocyte levels after HIV co-infection affect the integrity of granulomas, while other studies found no correlation, and some even suggested that HIV infection was unrelated to granuloma integrity. Although these studies agreed that HIV infection would lead to an increase in MTB load in infected individuals, the hypothesis that “HIV promotes MTB infection by reducing the formation of tuberculous granulomas” needs further verification. Here, we compare the pathological characteristics of tuberculous granulomas in biopsy specimens from patients with TB only and HIV-MTB co-infected patients, as well as the immunohistochemical distribution of three types of T lymphocytes (CD3+, CD4+, and CD8+), to explore the relationship and influencing factors between HIV infection and the formation of tuberculous granulomas.

## 2. Materials and Methods

### 2.1. Ethics

This research was approved by the Ethics Committee of Shanghai Public Health Clinical Center, Fudan University (protocol code: Public Health 2019-s030-02, Project number: Scientific Research Project of Shanghai Science and Technology Commission 18411970800). All patients with TB mono-infection and those co-infected with HIV and MTB were included in this study and signed a written informed consent form.

### 2.2. Study Subjects and Data Collection

The study subjects were patients with HIV-MTB co-infection treated at Shanghai Public Health Clinical Center from February 2017 to September 2020, and TB patients hospitalized from January to December 2020. The study collected patients’ clinical information relevant to the infection, including clinical manifestations, laboratory, radiological, pathological examination results, and clinical treatment information.

### 2.3. Specimen Collection

Patients underwent biopsies during hospitalization for clinical diagnosis and treatment, and the pathological specimens for this study were obtained from these biopsies. All biopsies were performed via fine needle aspiration. For patients co-infected with HIV and MTB, blood was collected during hospitalization according to routine clinical diagnostic and treatment procedures for peripheral blood CD4+ T lymphocyte testing.

### 2.4. Case Definition

The diagnosis of AIDS followed the standards in the Chinese Guidelines for Diagnosis and Treatment of Human Immunodeficiency Virus Infection/Acquired Immunodeficiency Syndrome (2021 edition) [10], and the diagnosis of TB followed the Guidelines for the Diagnosis and Treatment of Pulmonary TB (2001) [11]. The classification of TB aligned with the criteria outlined in TB classification WS196-2017 [12].

### 2.5. Histopathological Study

The biopsy specimens obtained by ultrasound-guided fine needle aspirate were routinely dehydrated, embedded in paraffin, cut into continuous sections 4–6 mm thick, and then stained with hematoxylin and eosin (HE staining), and subjected to special histochemical and immunohistochemical techniques.

### 2.6. Granuloma Morphology HE Staining Microscopy

Granuloma morphology HE staining microscopy was used to examine histopathological parameters such as inflammatory cell infiltration, granuloma formation, tissue necrosis, and fibrous tissue proliferation. Based on the pathology assessment, this study classified the integrity of granuloma formation into three categories: fully formed, incompletely formed, and unformed.

Fully formed granulomas included three types:Typical granulomas composed of epithelioid cells, multinucleated giant cells, macrophages, surrounding lymphocytes, and reactive proliferating fibroblasts forming a nodular structure with a clear boundary and no caseous necrosis at the center.Caseous necrotic granulomas (typical tuberculous granulomas) with complete coagulative necrosis (visually caseous necrosis) at the center of the granuloma.Fibrous granulomas mainly evolved from typical granulomas, predominantly composed of fibroblasts and fibrocytes, with a few epithelioid cells and macrophages [13];

Incomplete granulomas were defined as tissues observed under an optical microscope showing a trend towards granuloma formation but lacking a complete granuloma structure. All granulomas were assessed by a pathologist to confirm granuloma classification. Also, we used Image J (Version: 2.14.0/1.54f) to measure the area size of the fully formed and incompletely formed granulomas.

### 2.7. Immunohistochemical Tests

Immunohistochemical staining of paraffin tissue sections was performed using the kit containing following monoclonal primary antibodies according to standard operating procedures: CD3, CD4, CD8 (Product name: BD Multitest^TM^ CD3/CD8/CD45/CD4; Cat# 340499).

The standard process for immunohistochemical testing involved the use of a German Leica automatic immunohistochemistry machine, starting with Leica Bond-Max computer input and programming, and setting the procedure. The general process included deparaffinization, alcohol processing, antigen retrieval using EDTA or citrate salts (pH 6.0–9.0), and heat-induced epitope retrieval at 100 degrees Celsius for about 30 min using the machine’s in-built program, primary antibody incubation for about 20 min, and secondary antibody binding for about 20 min. Secondary antibodies and auxiliary reagents were purchased from the German company Leica Reagents (Wetzlar, Germany). Monoclonal primary antibodies were acquired from Shanghai Gene Technology Co., Ltd. (Shanghai, China). PBS was used as a blank control instead of the primary antibody, and known positive tissues were used as positive controls. DAB staining was followed by counterstaining with hematoxylin. After the Bond-Max machine signaled the end of the process, the software program was terminated. The slides were then dehydrated, cleared, and mounted with neutral balsam. The results were accurately determined based on the positive localization of different antibodies.

### 2.8. Quantitative Analysis of Immunohistochemistry

The material was evaluated with the use of a light microscope (Leica Reagents, Wetzlar, Germany)) using ×40 lenses. Then the digital document was processed using the ImageView (Version: 1.1.6.1024) software and further magnified by a factor of 10. Expression was calculated via the use of the Image J software (Version: 2.14.0/1.54f) IHC Profiler plugin. The scoring mechanism of this plugin has been described [14,15]. The percentages of high-positive and positive zones were viewed as the quantitative results of the CD3+, CD4+, and CD8+ T cells. Some blurred images were removed.

### 2.9. Statistical Analysis

Statistical analysis was performed using IBM SPSS Statistics 25, GraphPad Prism 8.0, and R×64 4.0.4 software. Comparisons between the two groups were conducted using *t*-tests, χ^2^ tests, and rank-sum tests. Logistic regression and Kendall’s tau-b correlation analysis were used to analyze factors related to the integrity of granulomas. A significant difference was set at *p* < 0.05.

## 3. Results

### 3.1. General Information

This study included a total of 102 patients, among whom 53 were co-infected with HIV/MTB and 49 were mono-infected with MTB. In the co-infection group and the mono-infection group, the proportion of male patients was 79.2% and 42.9%, respectively, with an average age of 41.40 (±13.74) years in the co-infection group and 27.71 (±20.04) years in the mono-infection group. In the co-infection group, 17 individuals were confirmed to have MTB infection through bacteriological examination, including species identification. The remaining 36 individuals were diagnosed based on positive acid-fast staining of pathological specimens, aspirate, sputum, urine, etc., and had a positive response to empirical anti-TB treatment. In the mono-infection group, 43 individuals were confirmed to have MTB infection through bacteriological examination. Additionally, six individuals were diagnosed based on positive acid-fast staining laboratory results and a positive response to empirical treatment. The species identification was usually based on the MTB IgG antibody test kit (colloidal gold method) and Xpert MTB/RIF. Most of the biopsy sites were extrapulmonary, including masses or lymph nodes in areas such as the mandible, neck, axilla, chest wall, abdomen, adrenal gland, kidney, groin, and arm. Specific details are presented in Table 1, Table 2 and Table 3. Furthermore, in the HIV-MTB co-infection group, there were five patients whose treatment regimens remained unknown. Among the remaining patients, before biopsy, 12 received no drug treatment, including antiviral treatment (ART) or treatment of TB; 1 received treatment against TB only; 15 received ART only; and the remaining 20 received treatment against both two diseases. In the MTB mono-infection group, excluding 2 patients whose treatment status remained unknown, 20 of the remaining patients had not received anti-TB treatment before biopsy.

### 3.2. Pathological Findings

#### 3.2.1. HIV Infection Reduced the Integrity of Granulomas

The pathological examination set after HE staining (Figure 1) revealed that in the HIV-MTB co-infection group, there were fully formed granulomas in 14 cases (26.4%), incompletely formed granulomas in 7 cases (13.2%), and no granuloma formation in 32 cases (60.4%). In the MTB mono-infection group, there were fully formed granulomas in 25 cases (51.0%), incompletely formed granulomas in 4 cases (8.2%), and no granuloma formation in 20 cases (40.8%). There was a statistically significant difference in the rate of fully formed granulomas between the two groups (difference 0.246, χ^2^ = 6.527, 95% CI 0.06–0.43, *p* = 0.011).

Additionally, we analyzed the size of the granulomas in both groups. The average granuloma area in the MTB mono-infection group was 2.02 mm^2^, while the average area in the HIV/MTB co-infection group was 2.52 mm^2^. The independent sample *t*-test showed that there was no statistically significant difference between the two groups in terms of granuloma size (*p* = 0.1869) (Figure 2).

#### 3.2.2. The Integrity of Tuberculous Granulomas in HIV-MTB Co-Infected Individuals Was Positively Correlated with Peripheral Blood CD4+ T Cell Count

Given that HIV infection can cause the destruction of CD4+ T lymphocytes, we conducted a statistical analysis of the absolute values of peripheral blood CD4+ T lymphocytes in both groups to investigate whether these values are correlated with the degree of granuloma completeness. Compared to the reference range of 410–1590/μL for healthy adults, the median level of peripheral blood CD4+ T lymphocytes in the HIV-MTB co-infection group was much lower at 78.5/μL (IQR = 153.75), while the median level in the MTB mono-infection group was 673/μL (IQR = 402). The Mann–Whitney test indicated that the level of peripheral blood CD4+ T lymphocytes in the HIV-MTB co-infection group was significantly lower than that in the MTB mono-infection group (Z = −6.07, *p* < 0.001).

In HIV-MTB co-infection patients, granulomas classified as “incompletely formed” and “unformed” were grouped together as “poorly formed granulomas” for analysis. A rank-sum test showed a statistically significant correlation between peripheral blood CD4+ T cell count and the formation and tendency of granulomas (*p* = 0.0018). Using Kendall’s tau-b correlation analysis to further explore the relationship between granuloma formation and peripheral blood CD4+ T lymphocyte count, a correlation coefficient of 0.49 was obtained, with *p* = 0.001 (Figure 3). Therefore, changes in the count of peripheral blood CD4+ T lymphocytes are positively correlated with the integrity of tuberculous granulomas, indicating that the infection status of HIV is consistent with the completeness of granuloma formation. However, the number of CD8+ T lymphocytes in the peripheral blood (R = 0.10, *p* = 0.419), the percentage of monocytes (R = −0.18, *p* = 0.134), and the percentage of neutrophils (R = −0.06, *p* = 0.583) were not related to the formation of granulomas. The original data are presented in Appendix A.

To further investigate other possible factors affecting the integrity of granulomas, we constructed a multifactorial logistic regression model considering factors such as gender and biopsy site (TB occurrence site) for the HIV-MTB co-infection group. The results of this model showed that, in addition to peripheral blood CD4+ T cell count, the biopsy site (*p* = 0.091) might be related to the integrity of tuberculous granulomas in HIV patients; no association was found between gender, peripheral blood monocyte ratio, and the integrity of granulomas (Table 4).

### 3.3. Immunohistochemistry Results

#### 3.3.1. HIV-MTB Co-Infection Leads to a Reduction in T Lymphocytes in Granulomatous Tissues

Immunohistochemical staining of biopsy tissues with CD3, CD4, and CD8 showed that compared to patients in the co-infection group, patients with MTB mono-infection had a denser population of T cells with positive immunohistochemical staining (Figure 4).

To compare whether there was a significant difference in the proportion of three types of lymphocytes between the two groups of patients, we further analyzed the proportion of CD3+, CD4+, and CD8+ T cells. We found that the proportions of CD4+, CD3+, and CD8+ T cells were significantly different between the MTB mono-infection group and the HIV/MTB co-infection group (Figure 5). Co-infection reduced the proportion of CD4+, CD3+, and CD8+ T cells in the granuloma tissues of patients, suggesting that HIV infection could lead to a decrease in the number of lymphocytes in tuberculous granuloma tissues.

#### 3.3.2. The Integrity of Tuberculous Granulomas in Co-Infected Patients Was Positively Correlated with the Proportion of CD4+ T Cells and CD3+ T Cells in Granulomatous Tissues

To determine whether the three types of lymphocytes could affect the integrity of granulomas, we conducted a semi-quantitative analysis of the completeness of granulomas and the presence of CD4+, CD3+, and CD8+ T cells in biopsy tissues. Through Kendall’s tau-b correlation analysis, we found a positive correlation between the proportion of CD4+ T cells in granuloma tissues and the integrity of the granulomas (correlation coefficient of 0.273, *p* = 0.032), indicating that a higher proportion of CD4+ T cells in the granulomatous tissue was more likely to result in complete granulomas. There was no relationship between CD3+ T cell proportion (correlation coefficient of 0.077, *p* = 0.542) and the integrity of granulomas. Also, the proportion of CD8+ T cells (correlation coefficient of −0.155, *p* = 0.217) showed no clear statistical correlation with whether granulomas were fully formed.

#### 3.3.3. The Count of Peripheral Blood CD4+ T Cells Was Positively Correlated with the Proportion of CD4+ T Cells in Granulomatous Tissues in Co-Infected Patients

Similarly, we conducted a regression analysis between the count of CD4+ T cells in peripheral blood and the proportion of CD4+ T cells in granulomatous tissues. The results showed a linear correlation between the two (*p* = 0.0293; regression coefficient: 0.02727; standard error: 0.01203) (Figure 6). However, the proportion of CD3+ T cells (*p* = 0.9799) and CD8+ T cells (*p* = 0.6408) did not show a linear relationship with the count of peripheral blood CD4+ T cells.

## 4. Discussion

Tuberculous granulomas are a typical structure resulting from MTB infection and composed of various immune cells, including neutrophils, macrophages, monocytes, CD4+ T lymphocytes, and CD8+ T lymphocytes. The granuloma structure has long been considered to limit the spread of bacteria, despite recent reports of its potentially detrimental roles [16,17]. The formation of granulomas is closely related to the body’s immune status, and it is believed that the varying clinical manifestations in TB patients result from a comprehensive histopathological response involving all granulomatous tissues [18]. The most significant outcome of HIV infection is the reduction in CD4+ T lymphocytes [6], and one of the signs of successful treatment is the recovery in the CD4+ T lymphocyte count, which is a marker of immune status [19]. This study explored the impact of HIV infection on the integrity of tuberculous granulomas and focused on the relationship between the quantity of CD4+ T lymphocytes in both granulomatous tissues and peripheral blood and the integrity of tuberculous granulomas.

Through HE staining, pathological analysis, and immunohistochemical analysis, we found that the count of CD4+ T cells in tissues linearly correlates with the integrity of granulomas. A higher CD4+ T cell count was more likely to result in fully formed granulomas, while a lower count may lead to poorly formed or unformed granulomas. According to immunohistochemical results, there was also a linear relationship between peripheral blood CD4+ T cells and CD4+ T cells in granulomatous tissues, and CD4+ T cells in granulomatous tissues decreased as peripheral blood CD4+ T cells decreased. Therefore, the peripheral blood CD4+ T cell count was associated with the integrity of granulomas, which was also confirmed in the initial logistic analysis of clinical data in this study. Other studies have shown that patients with a peripheral blood CD4+ T lymphocyte count greater than 300/mm^3^ would develop fully formed granulomas, those with counts between 200 and 300/mm^3^ could develop both fully formed and poorly formed granulomas, while those with counts less than 50/mm^3^ do not develop fully formed granulomas [20]. Our study further indicated that the number of peripheral blood CD4+ T lymphocytes positively affected the integrity of tuberculous granuloma formation. Therefore, the peripheral blood CD4+ T lymphocyte count can be used to predict the formation trend of tuberculous granulomas in HIV-MTB co-infected individuals, thus assessing the severity of the patient’s disease.

Our study’s findings regarding CD4+ T lymphocytes were consistent with previous clinical research results. Some studies have shown that a reduction in peripheral blood CD4+ T lymphocyte levels leads to decreased granuloma integrity [21,22]. For example, a 2016 clinical study in Africa suggested that reduced levels of peripheral blood CD4+ T lymphocytes could change the phenotype of immune cells within granulomas, thereby altering granuloma structure and morphology [23]. However, the mechanism by which HIV causes changes in the morphological structure of tuberculous granulomas is complex, and such research is mainly based on non-human primate models. Drawing from these studies, there are several possible reasons for the decrease in the integrity of tuberculous granulomas due to HIV infection. Firstly, HIV infection can cause dysfunction of macrophages and dendritic cells, reducing the innate immune system’s killing effect on MTB, altering the MTB-specific antigen presentation process and the initiation of adaptive immunity, leading to an increase in the host’s internal MTB load, accelerated infection progression, granuloma structure destruction, and a tendency for MTB to disseminate outside the lungs [24,25,26,27]. Secondly, in LTBI animal models, Simian Immunodeficiency Virus (SIV) infection usually leads to ATB, where SIV infection first depletes MTB-specific CD4+ T lymphocytes in granuloma tissues, and MTB-specific CD4+ T cells move less within granulomas [28]. Some clinical studies have proven that HIV co-infection reduces the number of MTB-specific CD4+ T cells in peripheral blood [29], lowers their proliferative capacity [30], and increases the intracellular viral load compared to non-specific T cells [31], and the function of MTB-specific CD4+ T cells was also long-term suppressed by HIV infection [32,33]. Additionally, some research suggests that SIV reduces the level of regulatory CD4+ T lymphocytes in the process of converting individuals with LTBI to ATB, thereby shifting the body’s immune response towards chronic inflammation, exacerbating damage [34]. Thirdly, in the host immune response caused by MTB infection, macrophages and CD4+ T cells expressing the chemokine receptor CCR5 [35,36] play a significant role, but these immune cells are also susceptible to HIV [37]. Therefore, granulomas formed after MTB infection contain a large number of cells that can facilitate HIV replication [38,39]. Meanwhile, in vitro experiments have found that tuberculous granulomas recruit a large number of immune cells expressing CCR5 [40], so the local HIV load in granulomas may increase, leading to the death of surrounding encapsulating cells and further compromising the integrity of the granuloma. Combining these research results, we believe that HIV infection would lead to a decrease in peripheral blood CD4+ T lymphocyte count, which in turn reduces the count of CD4+ T lymphocytes in tuberculous inflammatory lesions, with the latter affecting the morphology and integrity of tuberculous granulomas through various mechanisms.

The results of this study indicated that pathological examinations showed patients co-infected with HIV-MTB could lack typical granulomatous lesions [41], which was consistent with the clinical characteristic of co-infected patients being prone to disseminated infections. This conclusion was also supported by experiments in non-human primates [3]. Furthermore, the positive correlation between CD4+ T cell count in granulomas and the integrity of tuberculous granulomas explained, to some extent, the possible mechanisms behind the variation in clinical characteristics of HIV patients with different immune statuses when co-infected with TB. Co-infected individuals with different levels of CD4+ T lymphocyte counts present different clinical characteristics, i.e., the lower the CD4+ T lymphocyte count, the more likely TB is to manifest as extrapulmonary dissemination, lacking typical tuberculous granulomatous lesions and having different clinical features [42]. Combining these results, we believed that HIV infection affects the morphology and integrity of tuberculous granulomas, leading to different clinical characteristics in individuals with HIV-MTB co-infection compared to those with MTB mono-infection.

This study has its limitations: the sample size was small, so the results may not be fully representative; a detailed study of immune cells and immune status in the tissue was not conducted, and the local immune status was not investigated; most biopsy specimens in this study came from lymph nodes, with fewer from other tissues; and the order of infection with the two pathogens in the enrolled patients and the antiviral treatment plan were unknown, which may also have influenced the results.

Additionally, a recent study showed that co-infection with HIV can lead to changes in the immune cell composition of pulmonary tuberculosis granulomas, with an increase in CD68+ macrophages and a decrease in CD20+ B cells [43], while the results of this study showed that other immune cells in the peripheral blood were not statistically related to the integrity of the granuloma, which may be caused by the small sample size and the lack of research on other immune cells in the affected tissues.

In conclusion, through HE staining microscopy and immunohistochemical analysis of biopsy specimens from patients co-infected with HIV-MTB and those mono-infected with MTB, this study demonstrated that the level of peripheral blood CD4+ T lymphocytes is related to the integrity of tuberculous granulomas. This could provide research evidence to further elucidate the interrelationship between HIV and MTB in co-infected individuals and can also offer a reference for the clinical assessment and diagnosis of patients with HIV and concurrent TB.

## Figures and Tables

**Figure 1 viruses-16-01335-f001:**
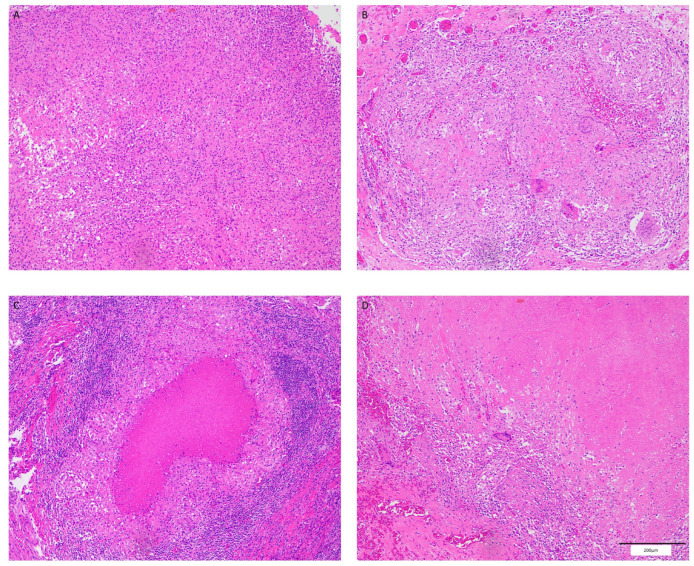
Different formation of granulomas (HE ×100): (**A**) incomplete granuloma; (**B**) typical granuloma; (**C**) caseous necrotic granuloma; (**D**) fibrous granuloma. Among the 4 pictures, (**B**–**D**) are all complete granulomas. Scale bar = 200 μm.

**Figure 2 viruses-16-01335-f002:**
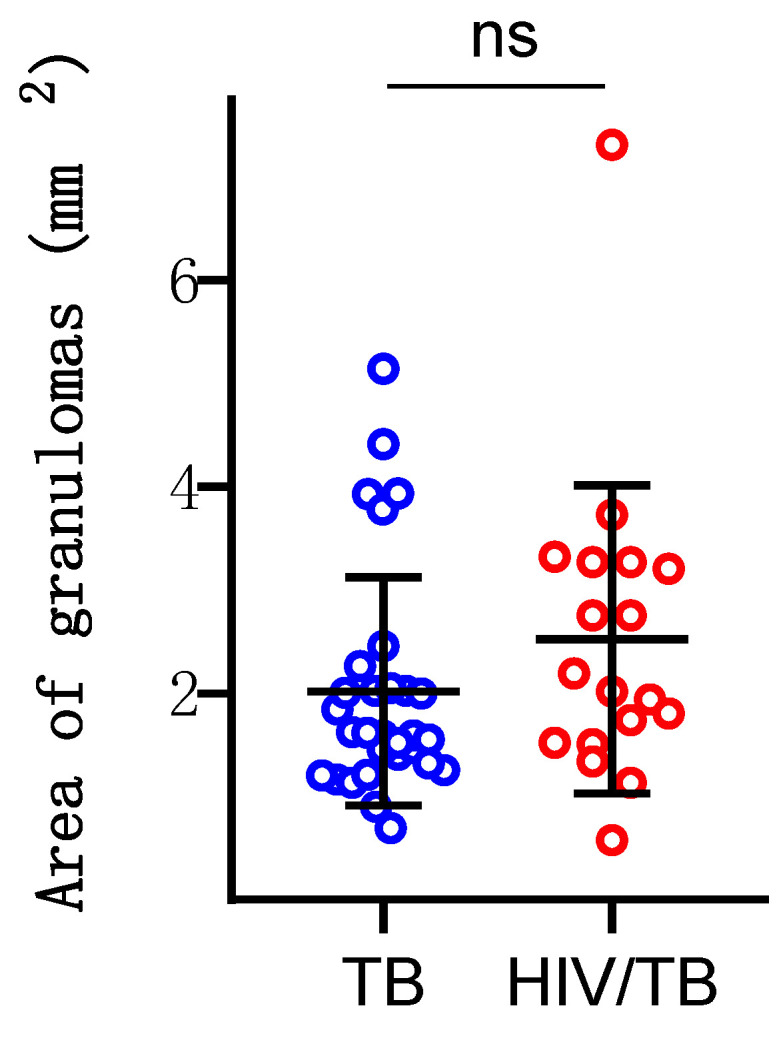
Area of granulomas in both groups.

**Figure 3 viruses-16-01335-f003:**
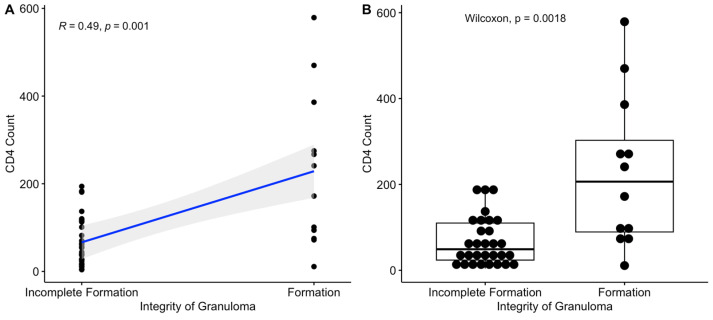
Statistical relationship between peripheral blood CD4+ T lymphocyte count and granuloma integrity in HIV-MTB co-infected patients: (**A**) Kendall’s tau-b correlation analysis between peripheral blood CD4+ T lymphocytes and granuloma integrity; (**B**) rank-sum test for the relationship between peripheral blood CD4+ T lymphocytes and granuloma integrity.

**Figure 4 viruses-16-01335-f004:**
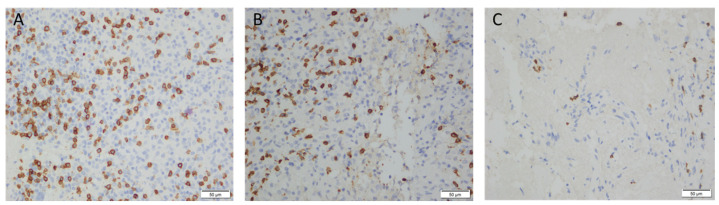
Immunohistochemical staining of granulomas (IHC ×400): (**A**) in a HIV-MTB co-infected patient’s lymph node biopsy tissue with granuloma formation, CD4 positive expression was seen in T cells, with the cell membrane appearing brown-yellow; (**B**) in a HIV-MTB co-infected patient’s colon tissue with proliferation of epithelioid cells and a tendency for granuloma formation, CD4 positive expression was seen in T cells, with cell membrane staining; (**C**) in a HIV-MTB co-infected patient’s lymph node biopsy tissue, complete coagulative necrosis was visible, with markedly fewer CD4 positive cells expressed. Scale bar = 50 μm.

**Figure 5 viruses-16-01335-f005:**
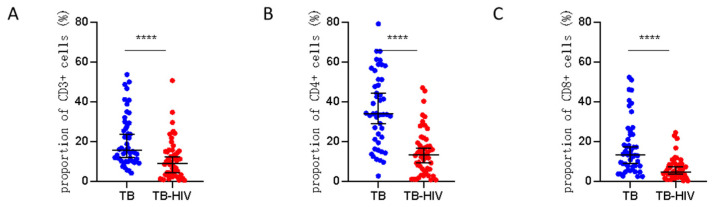
Statistics of positive cells after immunohistochemical staining: (**A**) proportion of CD3+ T cells in the tissue of patients in the HIV-MTB co-infection group and the MTB mono-infection group after immunohistochemical staining (t = 4.16, *p* < 0.001); (**B**) proportion of CD4+ T positive cells in the tissue of patients in the HIV-MTB co-infection group and the MTB mono-infection group after immunohistochemical staining (t = 7.02, *p* < 0.001); (**C**) proportion of CD8+ T cells in the tissue of patients in the HIV-MTB co-infection group and the MTB mono-infection group after immunohistochemical staining (t = 5.09, *p* < 0.001). **** stands for *p* < 0.0001 in Graphpad Prism.

**Figure 6 viruses-16-01335-f006:**
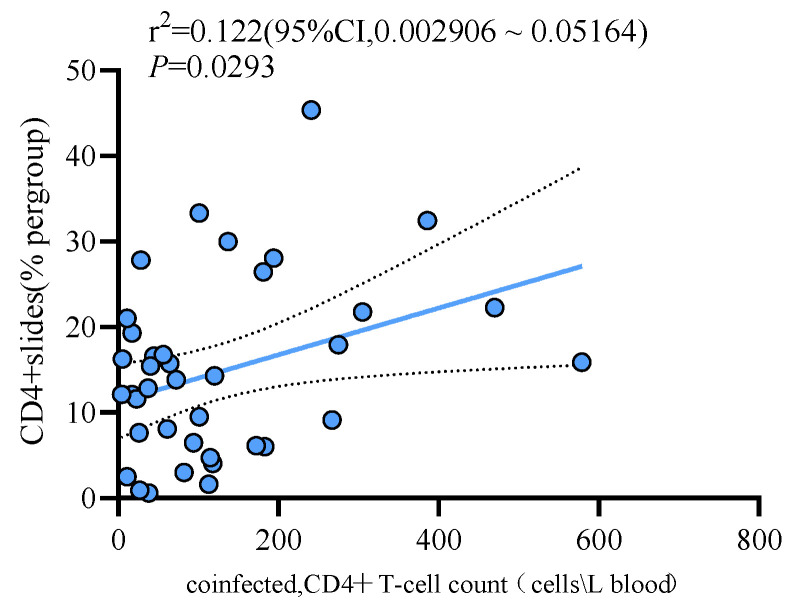
Linear correlation between peripheral blood CD4+ T cells and CD4+ T cells in biopsy tissues in patients of the HIV-MTB co-infection group.

**Table 1 viruses-16-01335-t001:** General information of the subjects.

Group	Average Age (Years)	Gender		Biopsy Site	
		Male	Female	Pulmonary	Extrapulmonary
HIV-MTB co-infection (*n* = 53)	41.40 (±13.74)	42	11	5	48
MTB mono-infection (*n* = 49)	27.71 (±20.04)	21	28	5	44

**Table 2 viruses-16-01335-t002:** Detailed information of the HIV-MTB co-infection group.

Subject No.	ART before Biopsy	Biopsy Site	Biopsy Anti-Fast Bacilli Staining	Treatment Regimens of TB before Biopsy
1	-	Neck (mass)	(−)	-
2	-	Neck (mass)	(+)	-
3	3TC + TDF + EFV	Neck (LN)	(+)	-
4	3TC + TDF + EFV	Neck (LN)	(+)	HRZE
5	-	Neck (LN)	(+)	-
6	3TC + TDF + EFV	Groin (LN)		-
7	-	Neck (mass)	(+)	-
8	-	Neck (mass)	(+)	-
9	-	Neck (mass)	(−)	-
10	3TC + TDF + EFV	Lung (mass)	(+)	-
11	3TC + TDF + EFV	Neck (LN)	(+)	-
12	-	Abdominal cavity (LN)	(+)	-
13	3TC + TDF + EFV	Abdominal cavity (LN)	(+)	HEZMfx + LZD
14	-	Mandible (LN)	(+)	HRZE
15	3TC + TDF + EFV	Neck (LN)	(+)	HRZELfx
16	3TC + TDF + EFV	Retroperitoneal (LN)	(+)	HRZE
17	3TC + TDF + EFV	Mediastinum (LN)	(+)	RfbLfx
18	No detailed regimen	Neck (LN)	(+)	-
19	3TC + TDF + LPV/r	Axilla (LN)	(+)	HREMfx
20	No detailed regimen	Adrenal gland	(−)	HRZE
21	3TC + EFV + LPV/r	Neck (mass)	(−)	RftEMfx
22	TDF + FTC + DTG	Lung (mass)	(+)	HRZE
23	3TC + EFV + LPV/r	Abdominal cavity (mass)	(+)	Mfx + CLI + LZD
24	FTC + DTG	Abdominal wall (mass)	(+)	HRZE
25	3TC + TDF + RAL	Abdominal cavity (mass)	(+)	-
26	3TC + TDF + EFV	Lung (mass)	(+)	-
27	TDF + FTC + DTG	Neck (LN)	(+)	HRZE
28	3TC + TDF + RAL	Mandible (LN)	(+)	-
29	TDF + FTC + DTG	Neck (LN)	(+)	HRfbZ
30	No detailed regimen	Neck (LN)	(+)	HREZLfx
31	3TC + TDF + EFV	Lung (mass)	(+)	HRZEMfx
32	-	Neck (mass)	(+)	-
33	3TC + TDF + EFV	Neck (LN)	(+)	HRZE
34	3TC + TDF + EFV	Neck (LN)	(+)	-
35	3TC + TDF + EFV	Abdominal cavity (mass)	(−)	-
36	3TC + TDF + EFV	Mandible (mass)	(+)	HRZELfx
37	3TC + TDF + EFV	Neck (mass)	(−)	-
38	3TC + TDF + EFV	Chest wall (mass)	(−)	RfbEPto
39	3TC + AZT + LPV/r	Neck (LN)	(+)	-
40	3TC + EFV + AZT	Neck (LN)	(+)	-
41	3TC + TDF + EFV	Pelvic cavity (mass)	(+)	RE
42	-	Posterior auricular LN	(+)	-
43	3TC + TDF + EFV	Lung (mass)	(+)	PaELfx
44	unknown	Kidney (mass)	(−)	unknown
45	unknown	Neck (mass)	(+)	unknown
46	unknown	Liver (mass)	(+)	unknown
47	-	Neck (mass)	(−)	-
48	3TC + EFV + ABC	Neck (LN)	(+)	-
49	-	Liver (mass)	(−)	-
50	-	Groin (LN)	(−)	-
51	unknown	Chest wall (mass)		unknown
52	TDF + FTC + DTG	Arm (mass)	(+)	-
53	unknown	Neck (LN)	(+)	unknown

“-” means the subject did not receive the treatment; “no detailed regimen” means the subject received the treatment, but the specifics remained unknown; “unknown” means there was no information about whether the subject received the treatment. Abbreviations: 3TC, lamivudine; TDF, tenofovir disoproxil fumarate; EFV, efavirenz; LPV/r, lopinavir/ritonavir; FTC, emtricitabine; DTG, dolutegravir; RAL, raltegravir; AZT, zidovudine; ABC, abacavir; H, isoniazid; R, rifampin; Z, pyrazinamide; E, ethambutol; Mfx, moxifloxacin; LZD, linezolid; Lfx, levofloxacin; Rfb, rifabutin; Rft, rifapentine; CLI, clindamycin; Pto, protionamide; Pa, pasiniazid.

**Table 3 viruses-16-01335-t003:** Detailed information of the MTB mono-infection group.

Subject No.	Biopsy Site	Biopsy Anti-Fast Bacilli Staining	Treatment Regimens of TB before Biopsy
1	Axilla (LN)	(+)	unknown
2	Axilla (mass)	(+)	-
3	Neck (LN)	(+)	-
4	Paraspinal (mass)	(−)	HRftELfx
5	Neck (LN)	(−)	HRZ
6	Neck (LN)	(−)	-
7	Neck (LN)	(+)	-
8	Neck (LN)	(+)	HRZE
9	Neck (LN)	(+)	-
10	Abdominal wall (mass)	(+)	HRZE
11	Chest wall (mass)	(+)	HRZ
12	Neck (LN)	(+)	-
13	Pleura	(+)	HRZE
14	Neck (LN)	(+)	HRZE
15	Chest wall (mass)	(−)	HRZE
16	Neck (LN)	(+)	-
17	Neck (LN)	(−)	-
18	Neck (LN)	(+)	-
19	Lung (mass)	(+)	HRZE
20	Pelvic cavity (mass)	(+)	HRZE
21	Neck (LN)	(−)	-
22	Neck (LN)	(−)	PaRftLfx
23	Neck (LN)	(+)	-
24	Neck (LN)	(−)	HRZMfx
25	Paraspinal (mass)	(+)	HRZE
26	Chest wall (mass)	(+)	HRZE
27	Neck (LN)	(+)	HRZE
28	Abdominal cavity (mass)	(−)	HRE + LZD + Am
29	Neck (LN)	(+)	HRZE
30	Axilla (mass)	(+)	HRZE
31	Neck (LN)	(+)	HRZE
32	Neck (LN)	(−)	HRZE
33	Psoas major muscle (mass)	(−)	HRZE
34	Neck (LN)	(+)	-
35	Neck (LN)	(−)	HRE
36	Neck (LN)	(−)	-
37	Chest wall (mass)	(+)	PAS + HRZ
38	Neck (LN)	(+)	HRZE
39	Neck (LN)	(−)	HRZE
40	Neck (LN)	(+)	-
41	Abdominal cavity (mass)	(+)	HRZMfx
42	Neck (LN)	(+)	-
43	Neck (LN)	(−)	-
44	Knee joint (mass)	(+)	-
45	Lung (mass)	(−)	unknown
46	Hip joint (mass)	(+)	-
47	Lung (mass)	(+)	HRZE
48	Neck (LN)	(+)	-
49	Axilla (mass)	(+)	-

“-” means the subject did not receive the treatment; “unknown” means there was no information about whether the subject received the treatment. Abbreviations: H, isoniazid; R, rifampin; Z, pyrazinamide; E, ethambutol; Mfx, moxifloxacin; LZD, linezolid; Lfx, levofloxacin; Rft, rifapentine; Pa, pasiniazid; PAS, para-aminosalicylic; Am, amikacin.

**Table 4 viruses-16-01335-t004:** Logistic regression analysis of factors influencing the formation of TB granuloma integrity in HIV-infected individuals.

Variable	Group	Wald	*p*	OR	95% CI for OR
Gender	Male				
	Female	2.125	0.145	7.848	0.492–125.245
Biopsy site	Pulmonary	2.852	0.091	0.129	0.012–1.389
	Extrapulmonary				
CD4+ T cell count in peripheral blood (/mm^3^)		4.493	0.034	0.995	0.991–1.000
Monocyte proportion in peripheral blood (%)		2.261	0.133	1.136	0.962–1.342

## Data Availability

The data presented in this study are available on request from the corresponding author.

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
