# Peer review of "HIV-MTB Co-Infection Reduces CD4+ T Cells and Affects Granuloma Integrity"

_viruses, 2024, doi:10.3390/v16081335_

Round 1

Reviewer 1 Report (New Reviewer)

Comments and Suggestions for Authors

Dear Authors,

I thank the authors for the effort to follow the recommendations, in order to improve the initial document.

I have provided my comments as follows:

#1 The title is too long. As such, I suggest that the authors carry out the reformulation and shortening to reflect the content of the study.

#2 The authors present a written manuscript, that requires an extensive editing of English language.

#3 The topic of the study is very interesting and not frequently reported.

#4 The literature is adequate, although I strongly recommend that more recent references should be included.

#5 The article is extremely confusing to read, it has numerous typos since the changes made remain in red. I request a more suitable version so that I can review it.

#6 It is imperative to add negative controls to each experiment!

Ex.: CD4+ T lymphocyte count in healthy individuals.

CD4+ T lymphocyte count in individuals with granulomas (not caused by HIV or tuberculosis infections).

Thank you.

Best regards.

Comments on the Quality of English Language

The article is extremely confusing to read, it has numerous typos since the changes made remain in red. I request a more suitable version so that I can review it.

The authors present a written manuscript, that requires an extensive editing of English language.

Author Response

#1 The title is too long. As such, I suggest that the authors carry out the reformulation and shortening to reflect the content of the study.

Reply: We agree with the reviewer’s suggestion. We have shortened the title to "HIV-MTB Co-Infection Reduces CD4+ T Cells and Affects Granuloma Integrity", which adequately reflects the main content of the article.

#2 The authors present a written manuscript, that requires an extensive editing of English language.

Reply: We appreciate the reviewer 1’s comments. We have polished the language of the entire text in the hope of making the content of the article easier to understand.

#3 The topic of the study is very interesting and not frequently reported.

Reply: We thank the reviewer 1’s appreciation. Indeed, the topic of this study is quite novel. Clinical samples with HIV and MTB co-infection are not easily obtained in clinical settings, resulting in limited research in this area. We hope that our findings, along with those of other researchers, will shed light on the close relationship between granuloma morphology and HIV and MTB infections, leading to an increase in related studies.

#4 The literature is adequate, although I strongly recommend that more recent references should be included.

Reply: We greatly appreciate the suggestion provided by reviewer 1. Through a literature review of recent years, we have added 6 more recent references to the discussion section.

#5 The article is extremely confusing to read, it has numerous typos since the changes made remain in red. I request a more suitable version so that I can review it.

Reply: We apologize for the grammatical lapses and typos of the manuscript and have carefully revised the manuscript and provided a clean version fore review.

#6 It is imperative to add negative controls to each experiment!

Ex.: CD4+ T lymphocyte count in healthy individuals.

CD4+ T lymphocyte count in individuals with granulomas (not caused by HIV or tuberculosis infections).

Reply: We thank the reviewer 1’s suggestion. The negative control in the experiment indeed plays a crucial role. While including healthy individuals as negative controls would certainly be beneficial, the samples used in this study are retrospective, and we did not collect additional blood and tissue samples from healthy individuals. Nevertheless, based on our clinical experience, we also provide a rough range of CD4+ T lymphocyte counts for healthy individuals as a reference (Ln 222-230). Meanwhile, the primary objective of this study is to compare various indicators between HIV-MTB co-infected patients and TB patients, who can serve as each other's controls without affecting the main conclusions of our article.

Thank you.

Best regards.

Reviewer 2 Report (New Reviewer)

Comments and Suggestions for Authors

HIV and tuberculosis co-infection reduces CD4+ T cells in peripheral blood and tissues and affects the integrity of granulomas

Dear author and editor:

The article talked about HIV – TB co infection and differences in granulomas integrity.

The article could be published after a minor revision .

 I have some comments on it:

·         The article should not be send in a revision modality.

·         According to your results could we say that HIV infection reduce the immune response against TB?

·         Is there any differences between the granuloma size between HIV and non-HIV TB infected patients.

·         In the results section, despite general information the author can say patients infection status or distribution.

·         Is there any differences between the percentage of the other immune cells CD8 , B-cells, monocytes , neutrophils, macrophages in the granulomatous formation.

Thank you very much, best regards.

Author Response

#1 The article should not be send in a revision modality.

Reply: As above, we agree with the reviewer 2’s suggestion. We have revised the entire text and provided a clean version for the reviewers.

#2 According to your results could we say that HIV infection reduce the immune response against TB?

Reply: Based on the current data, we believe that HIV can weaken the host's immune response to tuberculosis. For example, HIV infection significantly increases the incompleteness of tuberculous granulomas and reduces the number of T cells in peripheral blood. Of course, whether HIV has other inhibitory effects on the immune response to tuberculosis requires more data for support.We have corresponding discussions in the discussion section.

#3 Is there any differences between the granuloma size between HIV and non-HIV TB infected patients.

Reply: We appreciate reviewer 2’s insightful comments. Through quantitative analysis of granulomas in the two patient groups, we found that there was no significant difference in the size of the granulomas. These results are also included in the latest revised version(Ln 213-219).

#4 In the results section, despite general information the author can say patients infection status or distribution.

Reply: We thank reviewer 2’s comments and apologize for not including these details. Indeed, this information is helpful for understanding the article. However, it's unfortunate that most of the cases included in this study are outpatient cases, and much of the information is unavailable (since this information was not required at the time of patient admission). Therefore, we cannot provide these details, and we apologize for this once again.

#5 Is there any differences between the percentage of the other immune cells CD8 , B-cells, monocytes , neutrophils, macrophages in the granulomatous formation.

Reply: We appreciate the reviewer 2’s comments. Through statistical analysis, we did not find any correlation between the number of peripheral blood CD8+T lymphocytes,monocyte percentage, and neutrophil percentage in the granulomatous formation. This has been supplemented in Ln 240-243. Additionally, the number or percentage of B cells was not collected, so we couldn’t verify whether it was related to the granulomatous formation.

This manuscript is a resubmission of an earlier submission. The following is a list of the peer review reports and author responses from that submission.

Round 1

Reviewer 1 Report

Comments and Suggestions for Authors

Huang et. al. present data evaluating the correlation of peripheral CD4+ T lymphocytes and granuloma structure in TB patient and HIV/TB co-infected patients. The study addresses an important question regarding the role of tissue localization of CD4+ T lymphocytes on granuloma structure and function, but the data as presented is descriptive and has a number of limitations that prevent drawing conclusions on correlation of peripheral CD4+ T lymphocytes on tissue responses. Previous work cited by these authors (see DOI:10.1093/infdis/jiw313) provide a much more thorough assessment of LN granulomas as an example of how this manuscript could be improved. Furthermore, without knowledge of HIV treatment status (ART), I feel the conclusions cannot be supported.

Major comments:

·       Detailed information on study population needs to be provided in table format (age, sex, TB drug treatment, PPD status, drug treatment (TB drugs/antiretrovirals), etc. –should not say “data”

·       The location of the granulomas is not provided (pulmonary/extrapulmonary versus lymph node) until the discussion where it mentions, most granulomas were in LN (which LN?) and extrapulmonary—this is a significant limitation of the study and more information needs to be provided in the methods

o   Granulomas from different locations should be analyzed separately

o   How many granulomas were assessed per biopsy per patient? It is difficult to make conclusions about overall granuloma state within a patient if a single granuloma is characterized per patient.

·       Was TB diagnosis culture confirmed? PCR confirmed? Lineage?

·       Granuloma classification needs to include presence or absence of acid-fast bacteria and CFU

·       Semi-quantitative assessment of T cell numbers by IHC is not sufficient to address this question—was image analysis performed? No description of how the percentage of T cells was determined--cells/mm^2 needs to be reported and ideally a measure of bacterial burden

·       Were these granulomas assessed by a pathologist to confirm granuloma classification?

·       Images of granuloma classification need to show lower magnification in addition to high magnification to appreciate the differences

·       Flow cytometry assessment of PBMC and presence/magnitude of Mtb antigen-specific T cells (or some other measure of peripheral memory response would greatly add to this manuscript)

Minor comments:

·       What is meant by “positive localization of different clone number antibodies”

·       Title should reflect that this assessment was in LN granulomas and may not be generalizable to pulmonary granulomas

Comments on the Quality of English Language

Minor improvements in choose of phrase could improve readability. 

Author Response

The primary objective of this study was to investigate the impact of HIV infection on the integrity of tuberculoid granulomas and to analyze the relationship between CD4 lymphocyte counts in peripheral blood and tissue with the integrity of granulomas. The study findings suggest that HIV infection leads to a decrease in CD4 cell count, thereby affecting the integrity of tuberculoid granulomas. All patient diagnostic and treatment information has been supplemented in the revised manuscript. This research mainly focused on the relationship between HIV and granulomas; therefore, it did not involve the study of MTB bacterial load. Relatively few HIV/MTB co-infected patients were willing to undergo biopsy, which resulted in a relatively limited sample size for this study. With regard to tuberculosis, no differences have been found in the integrity of tuberculous granulomas at different locations, and considering this, the present study did not separately analyze biopsy tissues collected from different sites. The following is a point-by-point response to the reviewers' comments.

  1. Detailed information about patients has been added, including biopsy sites, but not PPD status, as no one in the co-infection group underwent the PPD test, and only half of the tuberculosis group did.

  1. Regarding the number of granulomas involved, each patient underwent fine needle aspiration biopsy, and the extracted tissue was sectioned for histological staining, so the specific numbers vary and are difficult to statistically recount now.

  1. Methods of tuberculosis confirmation have been supplemented to include bacteriological examinations such as acid-fast staining smears, cultures, and species identification. Patients not confirmed by bacteriological examination were clinically diagnosed with tuberculosis through histological staining/empirical treatment.

  1. Results of acid-fast staining of granulomas have been added (supplementary materials), without testing for bacterial load at the time of sampling, and the classification of granulomas in this article is limited to the morphology of granulomas and cannot be correlated with acid-fast staining results.

  1. Semi-quantitative cell counts for immunohistochemistry were captured and estimated by machine, detailed in the Materials and Methods section.

  1. Granuloma pathology images have been replaced, and classification results have been confirmed by a pathologist.

  1. Biological samples collected from the study subjects did not include blood cryopreservation, precluding experiments such as flow cytometry.

  1. The methodology of immunohistochemistry mentioned the use of monoclonal antibodies for cell staining, meaning the positive areas stained by CD3/CD4/CD8 monoclonal antibodies.

  1. Granulomas in this article did not solely originate from lymph nodes, and the title has not been modified as it was not restricted to pulmonary tuberculosis.

Once again, thank you for your instructive adive and warm help.

Yours sincerely,

Yinzhong Shen, MD, PhD

Shanghai Public Health Clinical Center

Fudan University

Reviewer 2 Report

Comments and Suggestions for Authors

Comments to the Authors of manuscript number: viruses-2837726 entitled “HIV and tuberculosis co-infection reduces CD4+ T cells in peripheral blood and tissues and affects the integrity of granulomas”.

This study investigates the impact of HIV infection on the formation and integrity of tuberculous granulomas in 102 tuberculosis patients. Biopsy specimens were collected from individuals with and without HIV co-infection. Through histological and immunohistochemical analysis, the study identifies factors influencing granuloma formation and explores the relationship between CD4+ T cells in peripheral blood and granulomatous tissue. The findings reveal that HIV infection is associated with poor granuloma formation, and there is a positive correlation between peripheral blood CD4+ T cell counts and the integrity of granulomas. Additionally, the study observes a positive correlation between the proportion of tissue CD4+ T cells and granuloma integrity. Overall, HIV infection appears to impact the morphology and structure of tuberculous granulomas, leading to reduced proportions of CD4+ T lymphocytes in both peripheral blood and tissue.

1. The introduction provides a comprehensive overview of the global impact of HIV and Mycobacterium tuberculosis, citing recent data from reputable sources such as the United Nations AIDS program (UNAIDS) and the World Health Organization (WHO). The statistics presented effectively highlight the significance of both pathogens and the scale of their impact on public health. The author appropriately emphasizes the complex interaction between HIV and MTB in co-infected individuals, addressing key aspects such as immunodeficiency, increased risk of tuberculosis progression, and the reciprocal influence on HIV replication. The rationale for the current study is clearly articulated, emphasizing the need for further research on the impact of co-infection on the integrity of tuberculosis granulomas. The study's objectives, comparing pathological characteristics and immunohistochemical distribution in biopsy specimens from patients with tuberculosis only and HIV/MTB co-infected patients, are logically derived from the identified research gap.

2. Material and methods is comprehensive and provides a clear understanding of the study's methodology, including patient selection, data collection, and laboratory procedures. The chronological order of steps enhances clarity.

3. Include a brief summary of the demographic characteristics of the study participants in material and methods. Information such as age, gender, and any other relevant demographic details could provide context and aid in the interpretation of results.

4. While the monoclonal antibodies used in immunohistochemical testing are mentioned, providing specific details such as antibody concentrations, sources, and clones used would add precision to the methodology.

5. Discussion successfully communicates the main findings of the study, emphasizing the relationship between peripheral blood CD4+ T lymphocytes and the integrity of tuberculous granulomas. The linear correlation observed is clearly presented.

6. The discussion adeptly connects the study's results with existing knowledge about the formation and function of tuberculous granulomas, as well as the impact of HIV infection on CD4+ T lymphocytes. This contextualization strengthens the significance of the study.

7. The study effectively discusses the clinical implications of the findings, such as the potential lack of typical granulomatous lesions in co-infected patients and the correlation between CD4+ T cell count in granulomas and clinical characteristics.

8. The limitations of the study are appropriately acknowledged, including the small sample size, lack of detailed immune cell studies, predominant use of lymph node biopsy specimens, and unknown order of infection and treatment plans. This transparency enhances the credibility of the study.

Round 2

Reviewer 1 Report

Comments and Suggestions for Authors

The authors failed to address the main methodological issue with this manuscript—that detailed correlation analysis is done based on subjective scoring by the authors for both granuloma morphology (H&E) and IHC staining. It is totally inappropriate to report “proportion of IHC positive cells” based on semi-quantitative scoring as in done in Figure 4.

The graphs should be changed so that the x-axis is score 1-4 based on the IHC scoring system they provided.

The subsequent correlation analyses are then based on the IHC scores and the granuloma scores. This is not robust. I feel this a fatal flaw of this manuscript and I would not report these findings unless the IHC analysis is done by quantitative image analysis which will require scanning the IHC slides and using a program such as Image J or HALO. This is not an unreasonable request since it does not require any additional experimentation—just slide scanning and image analysis.

Major comments:

Figure 4 misrepresents the data—The IHC is described as semiquantitative based on grading (Ln 146-152) with scores 1-4, then the IHC data is presented as a continuous variable  (y-axis listed as percentage). It is clear that one of the authors determined the percentage (by manual inspection since the Bond-Max cannot do this). This is a completely subjective assessment as reported in this manuscript. In clinical assessments—an H-score (semi-quantitative) can be measured but the results that are reported is the score (not the percentage). It is also typically done by two independent pathologists and then averaged for the final results. There is nothing wrong with semi-quantitative scoring even if done by a single person, but it needs to be clearly stated that this was the methodology.  All the results of this manuscript are therefore two sets of pathology scoring (H&E and IHC) which is not very robust. As mentioned in the previous review—the manuscript really needs quantitative image analysis (whole slide scanning and positive cells/mm^2) for the type of correlative analyses and results presented.

Minor comments:

Ln 54---this sentence does not make sense, rephrase

·       Suggest “trajectory of granulomas” or granuloma progression

·       “which known as local immune reponses”—needs to be reworded

Ln 82 –add reference to suppl figure for clinical information

Ln-97—earlier says FNA; here says puncture biopsy (please be consistent and use fine needle aspirate)

Ln 104—please replace “pathology readings” with pathological assessment

Ln 105—change to incompletely formed

Ln 137—remove this sentence  and discuss “immunoreactivity” or “immunopositivity” in the results section.

Ln 141—if the results are positive cells/mm^2 that is “quantitative” image analysis. Semi-quantitative is reserved for ordinal scoring done by a pathologist (such as H-score). The manuscript states that the ‘Bond-Max’ determined the positive staining and the cell counts, but this is an automated IHC staining machine. It does not do whole slide scans and cell counts. The paper describes a semi-quantitative scoring system (which was presumably done by the authors). Once again—semiquantitative scoring is NOT sufficient for correlative analyses for this type of question. The IHC slides for each Ab need to be scanned and quantitative image analysis (Image J) or some other mechanism needs to be used to quantify cells/mm^2.

Comments on the Quality of English Language

The authors do not have a good grasp of how to discuss the pathology data--I'm not sure if this is a language barrier or a lack of pathology training.